# Repurposing of Nitroxoline as an Alternative Primary Amoebic Meningoencephalitis Treatment

**DOI:** 10.3390/antibiotics12081280

**Published:** 2023-08-03

**Authors:** Javier Chao-Pellicer, Iñigo Arberas-Jiménez, Frieder Fuchs, Ines Sifaoui, José E. Piñero, Jacob Lorenzo-Morales, Patrick Scheid

**Affiliations:** 1Instituto Universitario de Enfermedades Tropicales y Salud Pública de Canarias, Universidad de La Laguna, Avda. Astrofísico Fco. Sánchez, S/N, 38203 San Cristóbal de La Laguna, Spain; alu0101016429@ull.edu.es (J.C.-P.); iarberas@ull.edu.es (I.A.-J.); isifaoui@ull.edu.es (I.S.); 2Departamento de Obstetricia y Ginecología, Pediatría, Medicina Preventiva y Salud Pública, Toxicología, Medicina Legal y Forense y Parasitología, Universidad de La Laguna, 38203 San Cristóbal de La Laguna, Spain; 3Centro de Investigación Biomédica en Red de Enfermedades Infecciosas (CIBERINFEC), Instituto de Salud Carlos III, 28220 Madrid, Spain; 4Department of Microbiology and Hospital Hygiene, Bundeswehr Central Hospital Koblenz, 56072 Koblenz, Germany; frieder.fuchs@uk-koeln.de; 5Institute for Medical Microbiology, Immunology and Hygiene, University Hospital Cologne, Faculty of Medicine, University of Cologne, 50935 Cologne, Germany; 6Parasitology Lab., Central Military Hospital Koblenz, 56072 Koblenz, Germany; 7Department of Biology, Working Group Parasitology and Infection Biology, University Koblenz, 56070 Koblenz, Germany

**Keywords:** *Naegleria fowleri*, primary amoebic meningoencephalitis, drug repurposing, Nitroxoline, programmed cell death, 8-hydroxy-quinoline

## Abstract

Among the pathogenic free-living amoebae (FLA), *Naegleria fowleri* is the etiological agent of a fatal disease known as primary amoebic meningoencephalitis (PAM). Once infection begins, the lesions generated in the central nervous system (CNS) result in the onset of symptoms leading to death in a short period of time. Currently, there is no standardized treatment against the infection, which, due to the high virulence of the parasite, results in a high case fatality rate (>97%). Therefore, it is essential to search for new therapeutic sources that can generate a rapid elimination of the parasite. In recent years, there have already been several successful examples of drug repurposing, such as Nitroxoline, for which, in addition to its known bioactive properties, anti-*Balamuthia* activity has recently been described. Following this approach, the anti-*Naegleria* activity of Nitroxoline was tested. Nitroxoline displayed low micromolar activity against two different strains of *N. fowleri* trophozoites (IC_50_ values of 1.63 ± 0.37 µM and 1.17 ± 0.21 µM) and against cyst stages (IC_50_ of 1.26 ± 0.42 μM). The potent anti-parasitic activity compared to the toxicity produced (selectivity index of 3.78 and 5.25, respectively) in murine macrophages and human cell lines (reported in previous studies), together with the induction of programmed cell death (PCD)-related events in *N. fowleri* make Nitroxoline a great candidate for an alternative PAM treatment.

## 1. Introduction

Naegleriasis is a global emerging disease in humans caused by the amoeboflagellate *Naegleria fowleri*. The disease is rare, but often fatal in humans [1]. The infection generally occurs in previously healthy children and young adults with a history of swimming and other recreational activities (e.g., diving, water skiing) in warm freshwater lakes and ponds. There are different risk factors, such as swimming, diving, or undertaking other recreational activities in nonchlorinated or poorly disinfected waters [2]. *Naegleria* amoebae are naturally found in humid soils and freshwater bodies but also in artificially heated waters, fountains, spas, and industrial cooling waters [3,4]. This parasite is usually found as trophozoites (vegetative stage) in water bodies. Moreover, in the absence of feeding sources, trophozoites turn into the flagellar stage, and if the environmental conditions become harsh, such as in the presence of disinfectants or extreme temperatures, the trophozoites turn into the cyst form [5]. Infection in humans usually starts with the inhalation of the trophozoites from a contaminated water source. However, the possibility of an infection via cyst-bearing dust has also been suggested, which transforms into its trophozoite form after penetrating the mucosa [6,7]. Once the epithelium is invaded by trophozoites, they migrate by attraction of certain cerebral neurotransmitters into the brain and meninges, leading to the term “brain eating amoebae” [8,9,10,11]. The clinical picture of this central nervous system disease presents with tissue damages as a diffuse meningoencephalitis, often with a fatal outcome [3,4]. The disease is therefore called Primary Amebic Meningoencephalitis (PAM) [12], *N. fowleri* being the only etiological agent of this disease. The striking feature of PAM is the rapid onset of symptoms following exposure to contaminated freshwater. Additional symptoms such as fever, headache, and vomiting may occur, which can rapidly progress to convulsions, hallucinations, and coma. Without any early treatment, the mortality rate is remarkably high [3,13,14].

The diagnosis of PAM is a major challenge, and it depends on the experience and knowledge of this pathogen and the resulting pathogenesis among clinicians and on the early and expedient identification of *N. fowleri*. A cerebrospinal fluid (CSF) sample from a lumbar punction is usually used for laboratory examination. After obtaining the wet CSF sample, three combined detection and determination methods are carried out: morphological microscopical analysis (light microscopy of the CSF sample directly and after cultivation), including induction assays to confirm different morphological stages and PCR [15].

As for the treatment of PAM, the therapeutic regimen currently applied consists of a combination of liposomal Amphotericin B and Miltefosine in synergy with other drugs, which have been confirmed to pass the blood–brain barrier (BBB) and eliminate the pathogen [16,17]. The therapy is usually combined with other substances, such as Dexamethasone or Mannitol, in order to reduce the intracranial pressure generated by the inflammation [18,19]. Despite this multidrug approach, the case fatality rate of PAM is still higher than 97%, and the side effects are associated with damage to major organs [15,20]. In the absence of a standardized clinical protocol, it is still necessary to search for novel drugs or drug regimens that act against different targets of the amoeba, leading to its elimination in the brain. The aim of the present study was to assess the potential of Nitroxoline for the treatment of the free-living amoeba *N. fowleri*. This antibiotic Nitroxoline is a quinoline derivate (5-nitro-8-hydroxyquinoline) that is structurally unrelated to any other antibiotic or antifungal substance and was first described as an antimicrobial in 1954 [21]. It has been rediscovered (respectively, repurposed) for the treatment of uncomplicated urinary tract infections (UTI, uUTI) and is recommended as a first-line drug in the German uUTI guidelines [22]. In 2016, the European Committee on Antimicrobial Susceptibility Testing (EUCAST) established a clinical breakpoint (classifying isolates as susceptible or resistant) for uUTI and *Escherichia coli*, and recently, many studies have shown promising activity in biofilms and against drug-resistant pathogens in vitro [23,24]. The mode of action is based on ion chelation with subsequent effects on enzymatic pathways and charges of cellular compartments of fungal and bacterial cells [25]. Recently, the activity of Nitroxoline against some emerging pathogens such as multidrug-resistant bacteria, including carbapenemase producers (enterobacterales), *Neisseria gonorrhoeae*, enterococci, *Escherichia coli*, staphylococci, and even drug-resistant fungi and mycobacteria, was demonstrated [23,26,27,28,29]. It has also been shown to be viricidal, being effective against Japanese encephalitis virus [30]. In the case of free-living amoebae, its activity against the pathogenic amoeba *Balamuthia mandrillaris* has been validated [31]. In 2023, Nitroxoline was successfully used in a patient in California, USA, with rare and usually fatal *Balamuthia mandrillaris* granulomatous amebic encephalitis (BAE). The patient survived after receiving treatment with Nitroxoline [32].

Laurie et al. [31] performed a scan of clinically approved compounds against the parasitic protozoan *Balamuthia mandrillaris*. In the study, Nitroxoline showed an inhibitory concentration IC_50_ of 4.77 μM against suspensions of *B. mandrillaris* trophozoites (amoebicidal activity). Preliminary studies on the in vitro efficacy of Nitroxoline diluted in dimethyl sulfoxide (DMSO) at different concentrations (range: 0.1 to 32.0 mg/L) against *Naegleria lovaniensis* resulted in activity values of 1.0 mg/L–8.0 mg/L. The work by Jochims et al. led to the conclusion that Nitroxoline should be further investigated as a promising candidate for treatment of *Naegleria* infections (Lit.: Jochims, K.; Fuchs, F.; Scheid, P.: Preliminary studies on the in vitro efficacy of the antibiotic nitroxoline in *Naegleria lovaniensis*, 30th Annual Meeting of the German Society for Parasitology (DGP 2023); abstract band and Poster presentation (2023)). This activity of Nitroxoline against different pathogens (bacteria, fungi, and parasites) and potent inhibition of biofilms in the era of antimicrobial resistance has recently led to regained interest in Nitroxoline repurposing.

The aim of the present study was to assess the potential of Nitroxoline for the treatment of another free-living amoeba besides *Balamuthia mandrillaris*: *Naegleria fowleri*.

## 2. Results

### 2.1. Nitroxoline’s In Vitro Activity against N. fowleri Trophozoites and Cytotoxicity

The in vitro activity of Nitroxoline against *N. fowleri* was evaluated using the alamarBlue^®^ assay, showing very similar IC_50_ values for both ATCC^®^ 30808™ and ATCC^®^ 30215™ of 1.63 ± 0.37 µM and 1.17 ± 0.21 µM, respectively (Figure 1A,B). Moreover, the assays against murine macrophages showed a CC_50_ value of 6.15 ± 0.52 µM (Figure 1C). Hence, the selectivity index value of Nitroxoline was 3.78 and 5.25 for both types of strains, indicating that the compound is selective in its action against *N. fowleri*.

### 2.2. In Vitro Cysticidal Activity of Nitroxoline against N. fowleri

The cysticidal activity of Nitroxoline against *N. fowleri* (ATCC^®^ 30808™) was determined (Figure 2). In general, the evaluated compound displays high values of population inhibition, with an IC_50_ of 1.26 ± 0.42 μM. In addition, despite being against the resistance form of the pathogen, the values exhibited are similar to those shown (described in the previous sub-section) against the trophozoite form of *N. fowleri* ATCC^®^ 30808™ (1.63 ± 0.37 µM). These values are still higher than those obtained for Miltefosine (21.52 ± 2.62 µM), which is a drug used in the first-line treatment of PAM.

### 2.3. Nitroxoline Induces PCD in N. fowleri Trophozoites

The evaluation of the type of cell death induced by Nitroxoline was performed using different commercial kits to expose some of the characteristic features of a programmed cell death (PCD). The experiments were conducted after the incubation of the *N. fowleri* ATCC^®^ 30808™ trophozoites with the IC_90_ (2.52 ± 0.47 µM) of the compound for 24 h. Amphotericin B and Miltefosine were used as positive control of PCD evaluation, the results of which were recently published by our research group [33].

#### 2.3.1. Chromatin Condensation

To evaluate if Nitroxoline induces chromatin condensation, the Hoechst 33342 stain (Life Technologies, Madrid, Spain) was used. This dye binds to the condensed chromatin and emits blue fluorescence at 461 nm. Furthermore, the propidium iodide (PI) (Life Technologies, Madrid, Spain) was also employed in this assay. The PI is able to cross the membrane of dead cells and bind to the nucleus, emitting a red fluorescence at 535/617 nm. In Figure 3, the blue fluorescence that corresponds to the condensed chromatin of treated cells (Figure 3D,E) is illustrated, whereas no fluorescence is emitted by the negative control (untreated) cells (Figure 3A,B). Regarding the red fluorescence of the PI, it is not visible in healthy trophozoites (Figure 3C), whereas Nitroxoline-treated amoebae show the red fluorescence of the PI-bound DNA (Figure 3F). These results suggest that the observed amoebae are undergoing a late apoptotic phase. The two-tailed *t*-test determined that the differences in the percentage of stained cells between the negative control and the treated cells are statistically significant in both experiments; *** *p* < 0.001 (Hoechst 33342), * *p* < 0.1.

#### 2.3.2. Plasmatic Membrane Damage

SYTOX™ Green dye was used to evaluate the damage caused by Nitroxoline at the plasmatic membrane level. This reagent, same as the PI, binds to nucleic acids and emits fluorescence. However, it is much more sensitive and effective in analyzing membrane permeability and binding to nucleic acids [34]. This probably explains the positive result obtained in the SYTOX^®^ Green assay (Figure 4), where high fluorescence was observed compared to the low level of fluorescence emitted by the PI (Figure 3). Negative control (untreated) amoebae (Figure 4A,B) are impermeable to the dye, and no fluorescence is emitted. On the contrary, Nitroxoline damaged the plasmatic membrane and the SYTOX™ Green entered the cell and bound to the nucleus, emitting an intense green fluorescence (Figure 4C,D).

#### 2.3.3. Mitochondrial Function Evaluation

The mitochondrial function was evaluated using two different assays. Firstly, the mitochondrial membrane potential was checked using the JC-1 Mitochondrial Membrane Potential Detection Kit (Cayman Chemicals Vitro SA, Madrid, Spain). This dye is presented forming J-aggregates and remains inside the mitochondria emitting red fluorescence when cells are in healthy conditions (Figure 5B). On the other hand, the JC-1 stain is released throughout the cell in its monomeric form at lower mitochondrial potentials. In Figure 5F, the green fluorescence emitted by the JC-1 stain when incubated with Nitroxoline-treated cells can be observed, while no green fluorescence is shown when incubating the dye with untreated cells (Figure 5C). Moreover, the ratio between the red and green fluorescence was calculated for both groups of cells. The two-tailed *t*-test performed determined that the ratio differences between the treated and negative control trophozoites are statistically significant (** *p* < 0.01).

Furthermore, the ATP levels were evaluated with the Cell Titer-Glo^®^ Luminescent Cell Viability Assay kit (Promega Biotech Ibérica, Madrid, Spain). Results are presented in Figure 6, where it is shown that the treatment of the trophozoites with the IC_90_ of Nitroxoline reduces the ATP production at 85% compared to the untreated cells (negative control). Moreover, the performed two-tailed *t*-test determines that this decrease is statistically significant (** *p* < 0.01), confirming the mitochondrial damage.

#### 2.3.4. Oxidative Stress Increase

To evaluate the oxidative stress in the amoebae (treated and non-treated), a fluorogenic probe, namely, CellRox, was used. This reagent, in the presence of diverse molecules, such as free radicals or peroxides, emits red fluorescence. For this reason, in viable amoebae (Figure 7A,B), where no oxidizing species were accumulated, no fluorescence was observed. In contrast, in Nitroxoline-treated amoebae, red fluorescence was observed due to the interaction of excessive ROS and the reagent (Figure 7C,D). Differences between the values were assessed using a two-tailed *t*-test. The results display significant differences (**** *p* < 0.0001) when comparing treated cells to the negative controls.

#### 2.3.5. Presence of Autophagy in Trophozoites

Autophagic processes were detected using MDC labelling (a fluorescent probe that accumulates in autophagic vacuoles [35,36] and analyzed via fluorescence microscopy. Figure 6C,D show that Monodansylcadaverine (MDC) stains autophagic compartments (brightest zones), which are indicative of a self-degradation pathway. Moreover, negative control amoebae (Figure 8A,B) display no fluorescence, indicating the absence of autophagic compartments.

## 3. Discussion

The antibiotic Nitroxoline is characterized by multiple bioactivities already described against several infectious agents [37,38]. This activity of Nitroxoline against different pathogens (bacteria, fungi, and parasites) and potent inhibition of biofilms in the era of antimicrobial resistance has recently led to regained interest in Nitroxoline repurposing. Laurie et al. (2018) [31] performed a scan of clinically approved compounds against the parasitic protozoon *Balamuthia mandrillaris*. Its mechanism of action has been previously discussed, suggesting that this amoebicidal activity may be based on the nitro group at position 5 and a hydroxyl group at position 8, corroborated by the loss of activity shown by its analogues. Hence, its activity, together with a number of key properties allowing it to cross the BBB, such as its low molecular weight (190.156 Da) and a high degree of lipid solubility [38], make this compound a candidate for the treatment of amoebic encephalitis. The aim of the present study was to assess the potential of Nitroxoline for the treatment of another Free-living amoeba (FLA) besides *Balamuthia mandrillaris*: *Naegleria fowleri*.

The in vitro assays against *N. fowleri* showed strong growth inhibition values against the tested trophozoite strains with IC_50_ values ranging from 1.17 to 1.63 µM. Moreover, Nitroxoline also showed cysticidal activity, with very similar values to the ones obtained for the vegetative stage. In addition, the selectivity index (CC_50_/IC_50_) demonstrated by Nitroxoline was 3.78, being higher than the one obtained for the reference drug to treat PAM, Miltefosine (3.30 for the ATCC^®^ 30808™ strain and 1.57 for the ATCC^®^ 30215™) [33]. Owing to its selectivity for *Naegleria*, programmed cell death (PCD) assays were conducted.

In order to be used as a therapeutic drug, the molecule must show the induction of an apoptotic-like death of the parasite. This type of PCD is predominant, producing diverse metabolic events in the cells that generate the death of the parasite in a regulated way. Additionally, recent studies have revealed new PCD pathways in protozoa, including cellular self-destruction (autophagy) pathways as a survival mechanism. On the other hand, a necrotic or deregulated cell death has been usually defined as premature cell death that occurs without molecular and morphological markers of apoptosis or autophagy [39,40]. The induction of this type of cell death could trigger inflammatory processes in which, together with the pathogenesis already generated by the disease, might lead to an early death of the patient. This death pathway can be avoided by the induction of the PCD process in the amoebae.

Therefore, the type of cell death is characterized by a series of typical morphological features and physiological markers in the amoeba. Consequently, one of the objectives of our study was to investigate the programmed cell death through the presence of characteristic metabolic events in order to destroy the pathogen without generating adverse damage in the patient. As described in the literature [41], PCD induction shows characteristic events such as chromatin condensation, alterations in plasma permeability, increased ROS production, decreased mitochondrial membrane potential, or the detection of autophagosomes.

After investigating the type of cell death evaluation, the induction of some cellular events compatible with the PCD were demonstrated by Nitroxoline in *N. fowleri*. The observed blue and red fluorescence after the treatment of the cells with the Hoechst and PI dyes indicate that the amoebae were undergoing later stages of apoptosis, a rapid action induced by Nitroxoline. In addition to these effects, an increase in oxidative stress (ROS overproduction) of the cell, plasma membrane permeability alterations, and severe mitochondrial damage (drastic decrease in ATP and ΔΨm levels) were also demonstrated. Furthermore, referring to previous studies [42,43], one of the main sources of ROS generation are the mitochondria. Based on this affirmation, the damage generated in this organelle could generate a pro-oxidant action, increasing the levels of oxygen radicals.

Another primary cell death pathway is autophagy, whereby cells degrade their own organelles via autophagosomes and lysosomes as a cell survival mechanism. This cell death pathway was observed in Nitroxoline-treated amoebae, as autophagic vacuoles were observed by binding to the MCD reagent. A similar process had already been described in cancer cells, in which Nitroxoline induced autophagy through an AMPK-dependent pathway [44]. In this study, it could be determined that the inhibition of Adenosine monophosphate-activated protein kinase (AMPKα), a factor that promotes autophagosome biosynthesis, reduced the drug-induced effects. Both processes exhibit different morphological and physiological characteristics; however, these pathways maintain a complex cooperation.

These results suggest that, through the events shown, amoebae are involved in an apoptotic and autophagic PCD process, most of them being associated with those already described for the first time in the genus *Naegleria* [41].

Our study has some limitations. Despite the clinical use of Nitroxoline for decades, pharmacokinetic data are limited to old studies with small sample sizes [37]. In particular, the idea that Nitroxoline may achieve relevant concentrations in the central nervous system has never been systematically investigated and is only based on some in vivo reports [32,45]. Therefore, the potential of Nitroxoline to cure PAM in vivo should not be overestimated based on in vitro findings, and more pharmacokinetic research is needed. However, the in vitro activity demonstrated within our study can be considered excellent, and the fact that Nitroxoline is an already approved infectious disease agent may facilitate the design of clinical trials or even compassionate use for PAM patients, as previously seen with an encephalitis patient suffering from *Balamuthia mandrillaris* [32].

## 4. Materials and Methods

### 4.1. Chemicals

Nitroxoline compound (Figure 9) was provided free of charge by Rosen Pharma St. Ingbert Germany. The stock solution was dissolved in dimethyl sulfoxide (Sigma Aldrich, Darmstadt, Germany) at a concentration of 40 mg/mL and preserved at −20 °C until required. Dilutions were prepared in Bactocasitone medium 2% (*v*/*w*).

### 4.2. Cellular Culture

In the present study, two clinical strains of *N. fowleri* (ATCC^®^ 30808™ and ATCC^®^ 30215™) were used to perform the in vitro assays, in which the activity of Nitroxoline (Rosen Pharma, Germany) was evaluated. Both *N. fowleri* strains used in this study were isolated from clinical samples, ATCC^®^ 30808™ (Strain KUL) being sourced from Belgium and ATCC^®^ 30215™ (Strain Nf69) collected in Australia. All *N. fowleri* strains were grown axenically in trophozoite form at 37 °C in Bactocasitone medium 2% (*v*/*w*) (Thermo Fisher Scientific, Madrid, Spain), supplemented with 10% (*v*/*v*) fetal bovine serum (FBS), 0.3% penicillin G sodium salt, and 0.5 mg/mL streptomycin sulphate (Sigma-Aldrich, Madrid, Spain). To induce the cyst stage, the trophozoites of *N. fowleri* (ATCC^®^ 30808™) were cultured axenically in MYAS medium at 28 °C. This liquid medium contains a high amount of salts which, together with slight agitation in an orbital shaker for ten days, provokes the encystment of the amoebae [46]. The strains were cultured in a biosafety level 3 (BSL-3) laboratory facility at the Instituto Universitario de Enfermedades Tropicales y Salud Pública de Canarias, University of La Laguna, as defined in Spanish basic guidelines in Real Decreto 664/1997 of 12 May 1997 on the protection of workers against risks related to exposure to biological agents at work.

For the cytotoxicity assays, a murine macrophages cell line J774. A1 (ATCC^®^ TIB-67) cultured in Dulbecco’s Modified Eagle’s medium (DMEM, *w*/*v*) supplemented with 10% (*v*/*v*) FBS and 10 μg/mL of gentamicin (Sigma-Aldrich, Madrid, Spain) was used. Cells were maintained at 37 °C and in a 5% CO_2_ atmosphere.

### 4.3. In Vitro Activity of Nitroxoline against N. fowleri

The evaluation of the in vitro activity of Nitroxoline (Rosen Pharma, St. Ingbert, Germany) against trophozoites of both *N. fowleri* strains (ATCC^®^ 30808™ and ATCC^®^ 30215™ seeded at 2 × 10^5^ cells/mL) was performed using a colorimetric assay based on the alamarBlue^®^ cell viability test to evaluate the inhibitory concentration 50 (IC_50_), as previously described [33]. Non-treated *N. fowleri* were used as negative control. Once the reagent was added (10% of the total volume), an oxidation-reduction process occurs, and the healthy cells that metabolize the reagent change their color from blue to pink. The fluorescence intensity was measured using the EnSpire^®^ Multimode Plate Reader (PerkinElmer, Madrid, Spain). The results obtained were analyzed in GraphPad Prism 9 software in order to obtain the inhibitory concentrations 50 (IC_50_) and 90 (IC_90_) for the subsequent study of PCD events.

After analyzing the activity of the molecule on trophozoites, the same assay was performed against the cyst stage. Once mature *N. fowleri* cysts were obtained, an activity plate was prepared using Nitroxoline (Rosen Pharma, Germany) in order to demonstrate its cysticidal activity. Therefore, serial dilutions of the substance in Bactocasitone were added to a 96-well microtiter plate (Thermo Fisher Scientific, Madrid, Spain). After dilution, the cysts of *Naegleria* were added at a concentration of 2 × 10^5^ cells/mL. After 24 h of cyst incubation, the growth medium was restored (fresh medium) in order to ease the excystation and determine the viability of the cysts. Non-treated *N. fowleri* cysts were used as negative control. Finally, alamarBlue^®^ cell viability reagent was added (10% of the total volume) to differentiate cysts and trophozoites on the plate was placed. Subsequently the plates were incubated for 72 h at 37 °C. The fluorescence was measured using the EnSpire^®^ Multimode Plate Reader (PerkinElmer, Madrid, Spain) and the IC_50_ was obtained from the analysis of the data.

### 4.4. In Vitro Cytotoxicity of Nitroxoline against Murine Macrophages

For this assay, a murine macrophage cell line J774A.1 (ATCC^®^ TIB-67) was used. Once the macrophages were incubated (10^5^ cells/mL) with the compound (mg/mL), the alamarBlue^®^ reagent was added (10% of the total volume), as previously described [47]. Non-treated murine macrophages were used as negative control. After 24 h of incubation, the fluorescence exhibited on the plate was analyzed using the EnSpire^®^ Multimode Plate Reader (PerkinElmer, Madrid, Spain). The data obtained were analyzed with GraphPad Prism 9 software to gain the cytotoxic concentration 50 (CC_50_).

### 4.5. Programmed Cell Death (PCD) Induction in N. fowleri

In the determination of the type of programmed cell death (PCD), *N. fowleri* ATCC^®^ 30808™ strain at a concentration of 5 × 10^5^ cells/mL was incubated. The induction of some metabolic events after the incubation of the pathogen with the IC_90_ of Nitroxoline (Rosen Pharma, Germany) for 24 h was observed. For this, different kits to detect the presence of this metabolic events were used following manufacturer’s recommendations. The results were revealed by the fluorescence captured using an EVOS M5000 fluorescence inverted microscope (Invitrogen by Thermo Fisher Scientific, Spain). For each objective lens thickness (40× and 100×), five images were taken. All assays were performed in triplicates.

#### 4.5.1. Detection of Chromatin Condensation

Induction of DNA condensation is one of the main characteristics of amoebae undergoing apoptosis. Therefore, a double-stain apoptosis detection kit (Hoechst 33342/PI) was used for this assay. Hence, the amoebae were incubated with Nitroxoline’s IC_90_ for 24 h at 37 °C. After the incubation time, Hoechst 33342 and propidium iodide (PI) were added at a certain concentration. Adding the reagents results in an intense blue fluorescence (excited at 350 nm and maximum emission at 461 nm) due to chromatin condensation in cells undergoing PCD and red fluorescence (excited at 535 nm and maximum emission at 615 nm) caused by the introduction of propidium iodide into dead cells.

#### 4.5.2. Analysis of Plasmatic Membrane Permeability

For the evaluation of plasma membrane integrity, SYTOX™ Green (Life Technologies, Madrid, Spain) reagent, which is impermeable to healthy cell membranes, was used. Following the incubation of *N. fowleri* with IC_90_ of Nitroxoline (Rosen Pharma, Germany) for 24 h, SYTOX™ Green dye was added at a final concentration of 1 μM for 15 min. The reagent uses the pores formed by the disruption of the plasma membrane to cross the plasmatic membrane. Once inside, it binds to nucleic acids, where it emits a 500-fold increase in green fluorescence.

#### 4.5.3. Oxidative Stress Evaluation

The oxidative stress induced in the amoebae was evaluated. For this purpose, the CellROX Deep Red fluorescent kit (Thermo Fisher Scientific, Madrid, Spain) was used. The assay was performed following the previous incubation of the amoebae with the IC_90_ of Nitroxoline (Rosen Pharma, Germany). Once treated, the test reagent was added at a final concentration of 5 μM for 30 min. Thereafter, the presence of reactive oxygen species (ROS) was evaluated via the detection of red fluorescence.

#### 4.5.4. Measurement of ATP Production

The Cell Titter-GLO^®^ Luminescent Cell Viability assay (Promega Biotech Ibérica, Madrid, Spain) was used to assess the ATP production in the amoebal trophozoites. Once the amoebae were incubated with the IC_90_ of Nitroxoline (Rosen Pharma, Germany) for 24 h, the same equivalent volume of Cell Titter-GLO^®^ was added. As result of the action caused by the molecule, production of ATP is interrupted by the transformation of luciferin to oxyluciferin. In the presence of oxyluciferin, the reagent will show ATP levels by luminescence proportional to ATP.

#### 4.5.5. Study of Mitochondrial Function Disruption

The change in the mitochondrial membrane potential of the amoebal trophozoites was analyzed using the JC-1 Mitochondrial Membrane Potential Detection Kit (Cayman Chemicals Vitro SA, Madrid, Spain). Following the incubation of the trophozoites with Nitroxoline (Rosen Pharma, Germany) for 24 h, 10 μL of JC-1 was added. Under usual conditions, the membrane potential is positively charged, causing the dye to form J-aggregates and emit red fluorescence (~590 nm). Once cell damage and, consequently, depolarization of this potential (ΔΨm) occur, the reagent disperses and remains as a monomer emitting green fluorescence (~529 nm).

#### 4.5.6. Autophagic Vacuole Tagging in *N. fowleri*

For the detection of autophagic processes, indicating another type of cell death besides apoptosis, the Monodansylcadaverine (MCD) reagent was used. In amoebae seeded with IC_90_ Nitroxoline (Rosen Pharma, Germany) for 24 h, the reagent was added. MCD, incorporated into multilamellar bodies via an ion-trapping mechanism and via interaction with membrane lipids, results in the observation of autophagic vacuoles. The cyan blue fluorescence can be observed using an EVOS M5000 fluorescence inverted microscope (Invitrogen by Thermo Fisher Scientific, Madrid, Spain) with an excitation wavelength of 335 nm and an emission wavelength of 525 nm.

### 4.6. Statistical Analysis

All experiments were performed in three independent assays, and the results were defined as the mean value ± standard deviation. The fluorescence data was obtained using EnSpire^®^ Multimode Plate Reader (PerkinElmer, Madrid, Spain). Once the values were obtained, data were analyzed using GraphPad Prism 9 by performing a nonlinear regression to obtain the amoebicidal activity, which will be represented as the IC_50_.

For the PCD study, IC_90_ was estimated from the IC_50_ and the Hill Slope obtained using GraphPad Prism 9. Moreover, quantitative analysis of the programmed cell death evaluation was carried out. The percentage of stained amoebae after the incubation of each evaluation kit with both treated and negative control cells was determined using the EVOS M5000 Cell Imaging System. The differences between the values are estimated using paired two-tailed *t*-test (compared with the negative control) also using GraphPad Prism 9 software. Results were considered statistically significant when the *p*-value was less than 0.05. For the JC-1 dye, the emitted fluorescence was measured, and the ratio between the red and green fluorescence was calculated. Experiments were carried out in triplicate, and each time, five different images were processed with a minimum of 80–100 cells. A two-tailed *t*-test was performed to evaluate the differences between both group of cells; * *p* < 0.1, ** *p* < 0.01, *** *p* < 0.001, **** *p* < 0.0001 were considered significant differences.

## 5. Conclusions

Nitroxoline showed activity at low micromolar concentrations against both *N. fowleri* strains. In addition, the activity exhibited against the stage with the highest tenacity (the cyst) was higher than the one exhibited against the trophozoite form of the ATCC^®^ 30808™ strain, a result that was remarkable because it was not expected. Finally, the type of cell death studies showed that this molecule induces a PCD process in the cells. Summing up, the antibiotic Nitroxoline could be repurposed as a therapeutic option for PAM treatment.

## Figures and Tables

**Figure 1 antibiotics-12-01280-f001:**
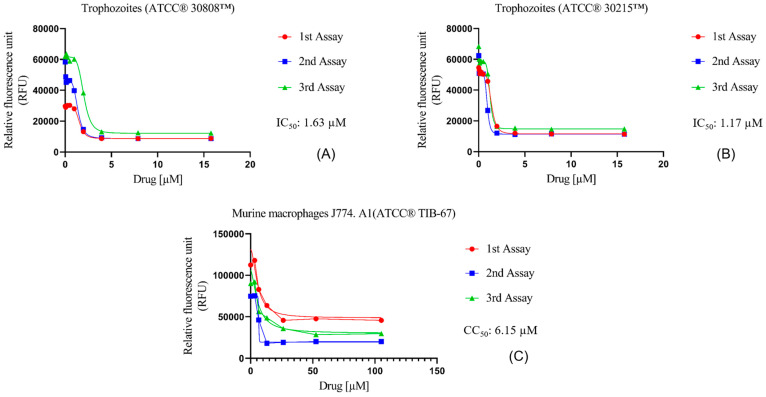
Dose–response curves of Nitroxoline against *N. fowleri* and murine macrophages J774A.1 (ATCC^®^ TIB-67). The amoebicidal activity of Nitroxoline against ATCC^®^ 30808™ (**A**) and ATCC^®^ 30215™ (**B**) strains of *N. fowleri* after 48 h is shown. In addition, the cytotoxicity effect against murine macrophages (**C**) was measured after 24 h of incubation with the drug. Activity and cytotoxicity experiments were performed using three independent assays, as illustrated in the graphical legends.

**Figure 2 antibiotics-12-01280-f002:**
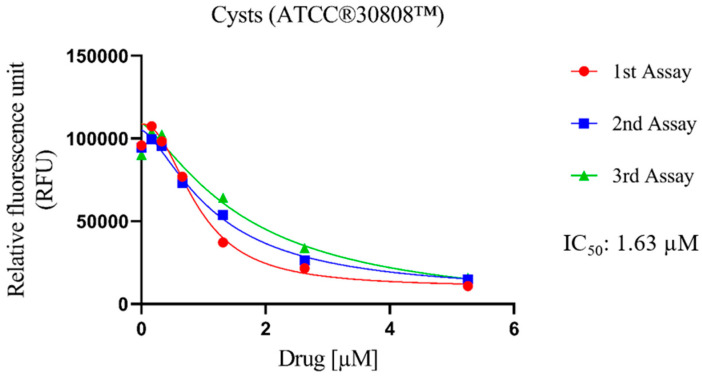
Dose–response curves of Nitroxoline against the cyst stage of *N. fowleri* (ATCC^®^ 30808™). This effect was measured by adding Nitroxoline for 24 h; then the drug was removed and replaced with fresh Bactocasitone medium for 72 h at 37 °C. The cysticidal activity experiment was performed using three independent assays, as illustrated in the graphical legends.

**Figure 3 antibiotics-12-01280-f003:**
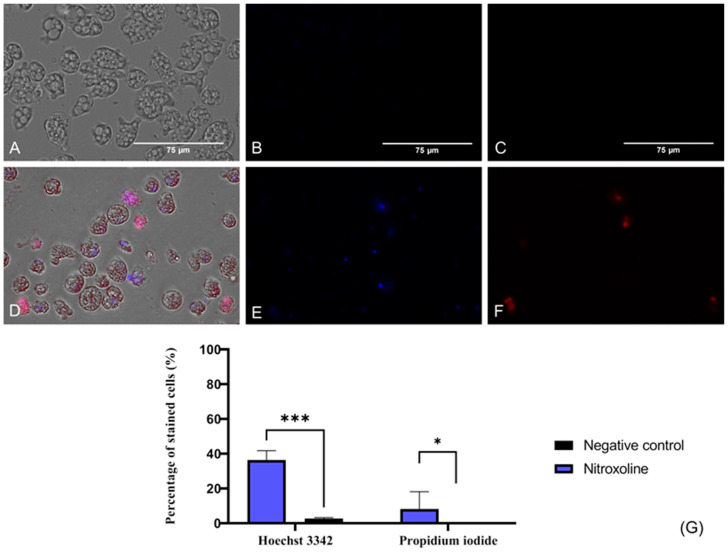
Presence of chromatin condensation using a double-stain assay apoptosis detection kit (Hoechst 33342/PI). *N. fowleri* trophozoites incubated with the IC_90_ of the Nitroxoline (**A**–**C**) for 24 h. Negative control (**D**–**F**). Treated cells show a blue fluorescence since the Hoechst 33342 binds to the condensed chromatin (**E**). The red fluorescence of the PI is also visible in treated cells (**F**). However, no fluorescence can be observed after the incubation of the Hoechst 33342 (**B**) and the PI (**C**) with the untreated cells. Images (×40) are representative of the cell population and were obtained with the EVOS™ M5000 Imaging System (Invitrogen by Thermo Fisher Scientific, Spain). Scale bar: 75 μm. (**G**) Quantitative analysis, the percentage of stained cells after the incubation of the treated and untreated trophozoites with the Hoechst 33342 and PI statins is represented in the bar graph. Data are shown as the mean value ± standard deviation (SD). The experiments were performed in triplicates. A two-tailed *t*-test was performed to determine the differences between the negative control and the treated cells; * *p* < 0.1; *** *p* < 0.001.

**Figure 4 antibiotics-12-01280-f004:**
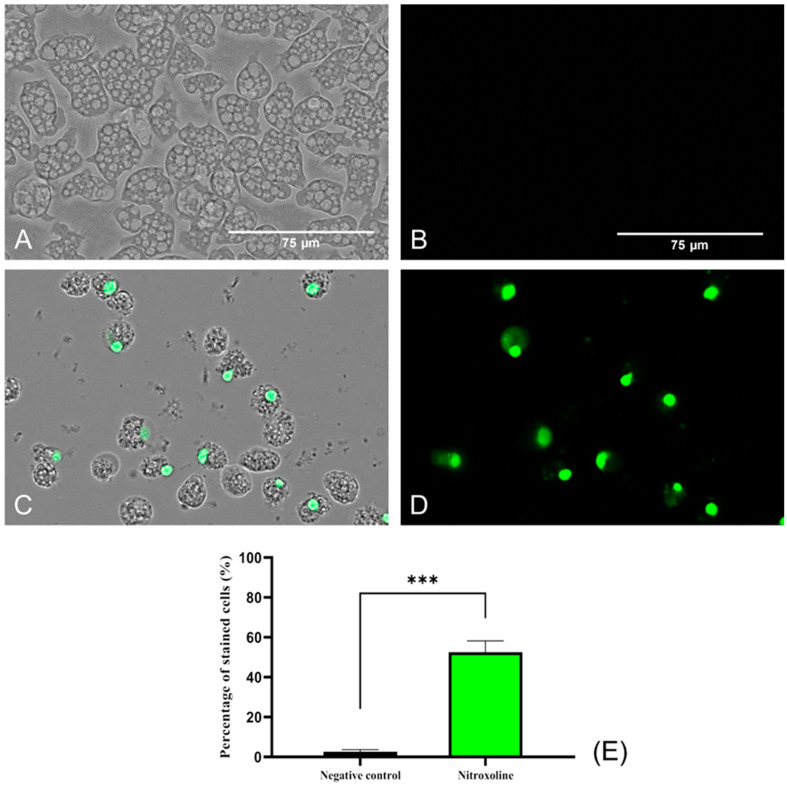
Plasma membrane permeability assay with the SYTOX™ Green dye. Nitroxoline IC_90_-treated trophozoites during 24 h (**C**,**D**) show an intense green fluorescence after the binding of the stain to the DNA. On the other hand, untreated amoebae (**A**,**B**) show no fluorescence since the dye is not capable of entering to the cytoplasm and hence cannot bind to the nucleic acids. Images (×40) were taken with the EVOS™ M5000 Imaging System (Invitrogen by Thermo Fisher Scientific, Spain) and are representative of the cell population. Scale bar: 75 μm. (**E**) The graph shows the percentage of stained cells of treated and negative control cells, represented as mean value ± SD. Three independent assays were conducted with the EVOS M5000 software tools. Five different images were processed each time. The differences between the values of both groups of amoebae were determined using a two-tailed *t*-test; *** *p* < 0.001.

**Figure 5 antibiotics-12-01280-f005:**
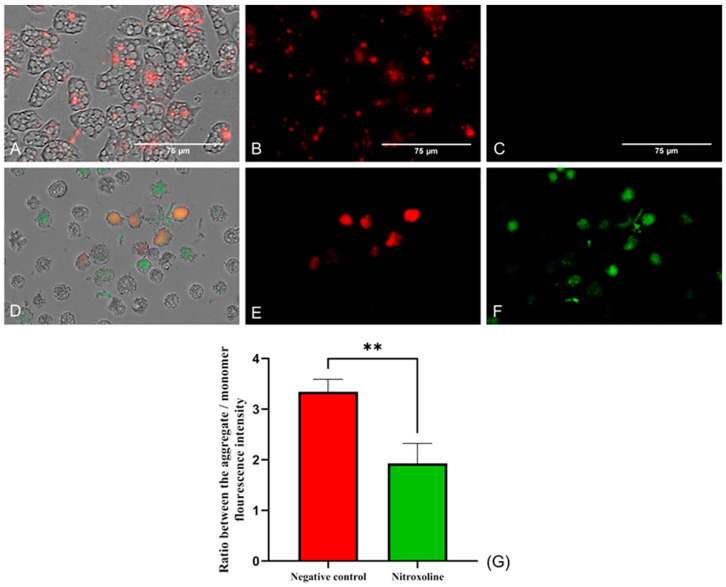
JC-1 Mitochondrial Membrane Potential Detection Kit assay. Cells incubated with the IC_90_ of Nitroxoline for 24 h (**D**–**F**). Untreated cells (**A**–**C**). In healthy conditions, the stain remains in the mitochondria forming J-aggregates and emits a red fluorescence at 590 nm (**B**). When the mitochondrial membrane potential is decreased, the dye is presented in monomers and emits a green fluorescence at 529 nm (**F**), while the red fluorescence is reduced (**E**). No green fluorescence can be seen in the negative control (**C**). Images (×40) are representative of the cell population in well and were taken in the EVOS™ M5000 Imaging System (Invitrogen by Thermo Fisher Scientific, Spain). Scale bar: 75 µm. (**G**) The graph shows the ratio between the fluorescence emitted by the aggregate and the monomer forms of the JC-1 after the incubation with the treated and negative control cells. Data are presented as mean ± SD after three independent assays. The fluorescence was evaluated using the EVOS M5000 software tools, and five images were analyzed each time. The statistical difference between the ratios of both group of cells was determined by a two-tailed *t*-test; ** *p* < 0.01.

**Figure 6 antibiotics-12-01280-f006:**
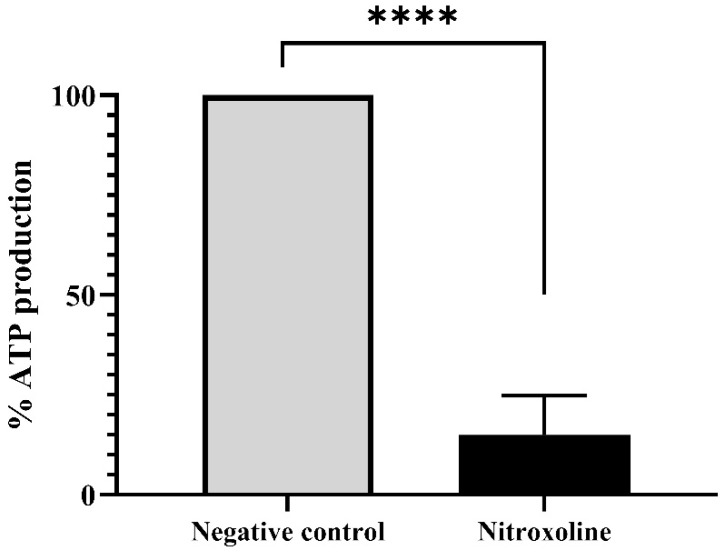
ATP production levels of Nitroxoline-treated cells in comparison to the untreated cells. Treated cells were incubated with the IC90 of Nitroxoline during 24 h. Data are presented as mean value ± SD. Three independent assays were carried out. Nitroxoline reduced the ATP production of the cells an 85% compared to the negative control. A two-tailed *t*-test was performed to determine the difference between the ATP production of both group of cells (**** *p* < 0.0001).

**Figure 7 antibiotics-12-01280-f007:**
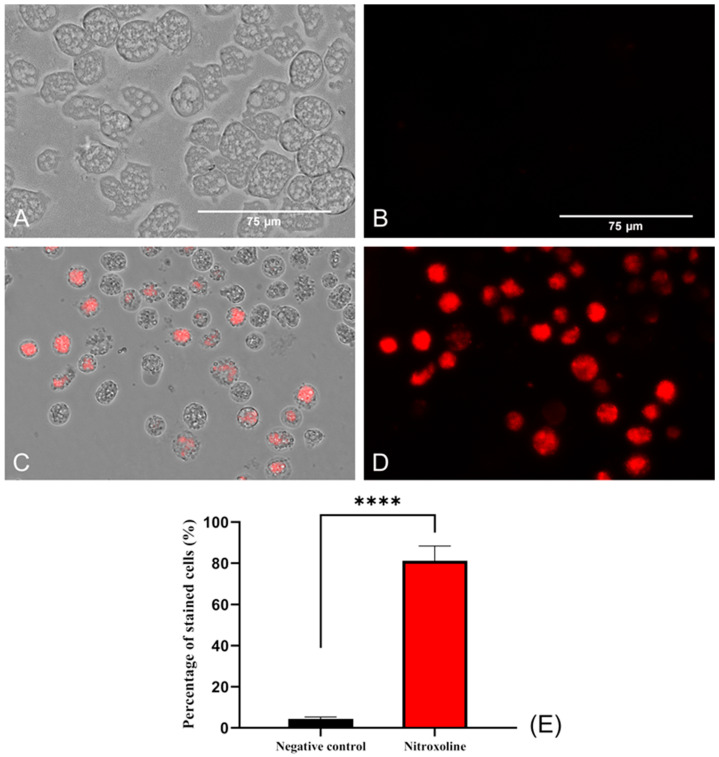
Oxidative stress assay with CellROX™ Deep Red. Nitroxoline IC_90_-treated trophozoites during 24 h (**C**,**D**) show an intense red fluorescence after the interaction of reagent with ROS overproduction. On the other hand, non-treated amoebae (**A**,**B**) show no fluorescence due to the absence of ROS accumulation. Images (×40) are representative of the cell population and were obtained using the EVOS™ M5000 Imaging System (Invitrogen by Thermo Fisher Scientific, Spain). Scale bar: 75 μm. (**E**) The graph shows the percentage of stained cells of treated and negative control cells, represented as mean value ± SD. Three independent assays were conducted using the EVOS M5000 software tools. Five different images were processed each time. The differences between the values of both groups of amoebae were determined using a two-tailed *t*-test; **** *p* < 0.0001.

**Figure 8 antibiotics-12-01280-f008:**
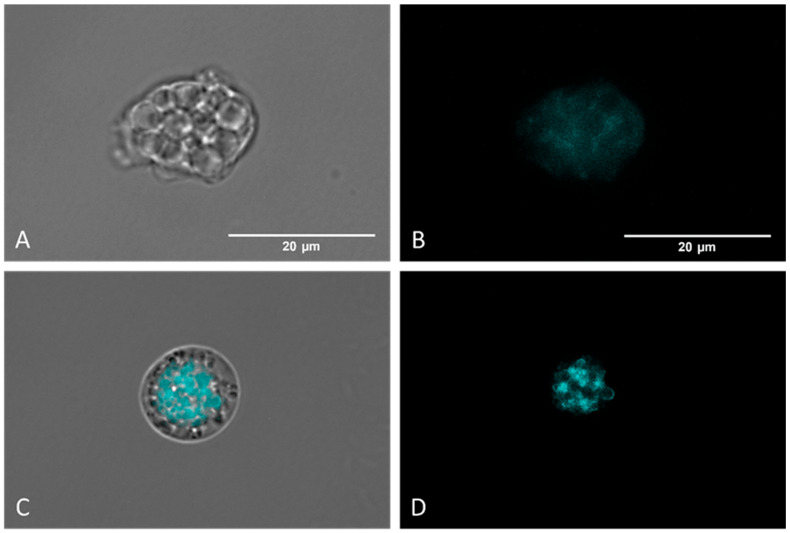
Detection of autophagic vacuoles via MCD labelling. Controls (untreated) (**A**,**B**) present poor and diffuse labelling. Nitroxoline-treated parasites (**C**,**D**) display strongly colored areas, indicating an accumulation of autophagic compartments. Images (×100) were taken with the EVOS™ M5000 Imaging System (Invitrogen by Thermo Fisher Scientific, Spain) and are representative of the cell population. Scale bar: 20 μm.

**Figure 9 antibiotics-12-01280-f009:**
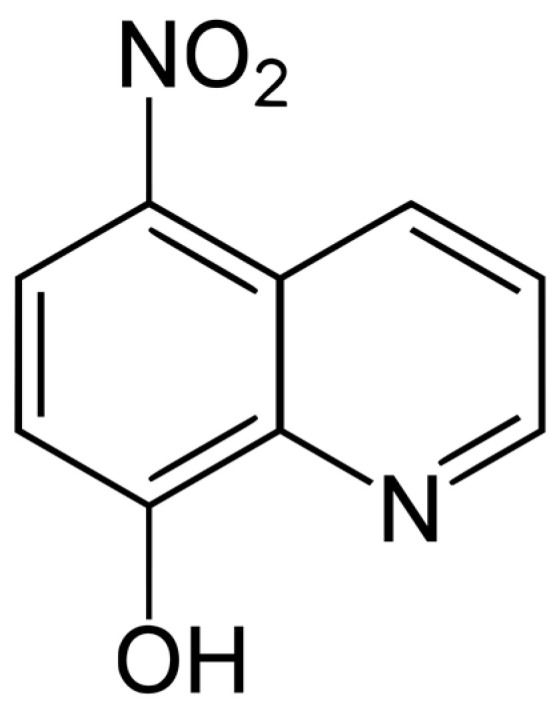
Nitroxoline chemical structure.

## Data Availability

Data are available upon request.

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
