# Peer review of "Repurposing of Nitroxoline as an Alternative Primary Amoebic Meningoencephalitis Treatment"

_antibiotics, 2023, doi:10.3390/antibiotics12081280_

Round 1
Reviewer 1 Report
1. The abstract needs to be revised by showing research results in the form of numeric and significance values (p-values).
2. The introduction is enough to make 4/5 coherent paragraphs.
3. There are several writings of the Latin name "N. fowleri" which have not been italicized; please italicize it!
3. All figures quality need to be improved; because the current format is still far from meeting the standards for publication. The graph should be enlarged to show more visible bars.
4. Provide a special conclusion section without being combined with the discussion; to make it easier for the reader.
Moderate editing of English language (grammar and spelling) required
Reviewer 2 Report
The manuscript "Repurposing of Nitroxoline as an anti-Naegleria compound" presents relevant results on the use of the antibiotic Nitroxoline as a proposal for the treatment of amoebic meningoencephalitis caused by Naegleria fowleri.
Below are my considerations:
- In line 96, studies performed in vitro are cited that demonstrated the effectiveness of the antibiotic in treating Naegleria lovanensis. I suggest mentioning the main techniques and results found in this cited work.
- The beginning of the discussion of the results (Lines 258-260) presents information that is repeated in the introduction of the paper.
- Regarding the autophagy process in lines 313-314 how does this process occur with cancer cells? Is it similar to the process described? Clarify.
- In materials and methods mention the origin of the strains.
- What would be the additional tests needed to be performed for the implementation of nitroxoline treatment for amoebic meningoencephalitis with Naegleria fowleri as the etiologic agent. Also, since the drug is not yet used, what would be the possible drawbacks.
- Line 96 Neagleria lovanensis check the spelling of the name.
Reviewer 3 Report
Title: Repurposing of Nitroxoline as an anti-Naegleria compound
This manuscript is not ready to publish in the journal as many weak points were presented in it. However, I do believe that if they can improve the manuscripts following all comments. It might have a chance to publish in the journal.
Comments
1. Topic: The title is not clear and should be modified.
2. Abstract is an overall talking about the research. Abstract contains a short sentence of each introduction, objective, and method. The authors should explain the interesting results followed by the conclusion. Hence, the authors have to summarize the story of the research, and describe in the abstract. I strongly suggest the authors to modify or re-write the abstract.
3. Line 33-34: This paragraph is too short. Please check.
4. Line 41: The full name of Naegleria fowleri is presented for the first time. The next time, it should be N. fowleri. Please correct in the whole manuscript.
5. In this manuscript, there are 13 paragraphs in introduction. It is too much for the readers to understand the point. Actually, 3-4 paragraphs are suitable. I strongly suggest the authors to modify or re-write the introduction.
-The first paragraph: describe the importance of N. fowleri in human
-The second paragraph: describe the treatment of N. fowleri infection, and give the information of Nitroxoline
-The third paragraph: describe the failed treatment of N. fowleri infection by other antibiotics as well as Nitroxoline
-The fourth paragraph: describe the objective of this study. Why the authors are interested in this study?
Results and discussion
6. Line 130: Delete “(see Material and Methods section)”
7. Figure 1G and Figure 2E: The information in Y-bar (Percentage of stained cells) should be replaced by “Percentage of death cells”
8. In the discussion, the 1st, 2nd, and 3rd paragraphs should be combined and re-written.
9. Line 274-281: Most of the sentences are repeated the results.
10. In the discussion, the authors should explain the mechanisms of action of Nitroxoline on Naegleria fowleri. What did you find? Which one of your findings is the novelty?
11. Please add the information of the discussion. Try to compare the results (the author’s hypothesis) with other finding by other researchers.
Materials and methods
12. Line 331-332: In the present study, two clinical strains of Naegleria fowleri (ATCC® 30808™ and ATCC® 30215™): Actually, ATCC is the reference strains. Please check and correct.
13. Line 330-349: The authors used several paragraphs, please reduce the numbers of paragraphs.
14. Line 338: The authors used the cysts of N. fowleri. Please describe the information of cyst in the introduction.
15. Actually, N. fowleri is the microorganism in the water. The suitable temperature is 25-30 degree Celsius. Why does the parasite is incubated at 37 (for the trophozoite) and 28 (for the cyst) degree Celsius?
16. “4.2 In vitro activity of Nitroxoline against Naegleria fowleri” and “4.3 In vitro cytotoxicity of Nitroxoline against murine macrophages”, please add more detail on the concentration of the parasites and the concentration of the antimicrobial agent used in this study.
17. “4.4 In vitro activity of Nitroxoline against cyst stage of Naegleria fowleri” is similar to item 4.2. Please combine the information in the same item.
18. Line 440: N. fowleri should be written in italic.
19. The references of 2022,2021, and 2023 are suggested to be cited.
20. Please remove some old references.
-
Reviewer 4 Report
Naegleria fowleri is a free-living amoeba that infects the central nervous system and causes the usually fatal disease, primary amoebic meningoencephalitis (PAM). Treatment includes a combination of drugs including amphotericin B, azithromycin, rifampin, dexamethasone, fluconazole, and miltefosine but these have poor efficacy and toxicity. The current drug options have been chosen empirically and better drugs with improved selectivity and efficacy are urgently needed. This manuscript explores an interesting idea of repurposing the approved antibiotic and cancer drug, nitroxoline for PAM. The authors explore various fluorescence readouts to evaluate in vitro activity of nitroxoline against Naegleria fowleri. However, the manuscript does not provide sufficient details about the data, methods, controls, and statistics to make conclusions about cidal anti-parasitic activity and selectivity of nitroxoline. The data in the current format rather suggests a lack of selective activity against the parasite. Please see below for more detailed comments.
Specific comments:
§ Abstract does not include sufficient details about the study. Additionally, the conclusions of the abstract appear different from rest of the manuscript as the abstract suggests nitroxoline is a good tool compound for studying PCD.
§ Introduction:
o Most of the manuscript focuses on PCD, but there is no background on it in the introduction. Instead, lines 282 to 292 of the discussion section highlight the rationale for following-up on in vitro activity by evaluating programmed cell death (PCD). Herein, it is mentioned that “the molecule must show the induction of an apoptotic-like death of the parasite”, and an alternate necrotic or degraded cell-death could lead to an early patient death. Could the authors kindly provide references to these claims for PAM patients. The PCD assays lack use of existing drugs as controls to evaluate how nitroxoline compares to them.
o Kindly introduce the cyst stage and the relevance of determining compound activity against this stage.
o Line 78: Expand EUCAST; and unclear what is meant by “established a clinical breakpoint”, please clarify.
o Line 85-87: Nitroxoline was shown to be active against JEV, and not the whole Flavivirus family in the reference cites, kindly clarify the statement.
o Lines 69-72 and 105-106 are redundant on the aim of the study. Please also include a summary of the studies done at the end of the introduction.
§ Include a chemical structure of nitroxoline.
§ For the in vitro activity against N. fowleri, there is only a mention of IC50 for both strains and IC90 for only ATCC® 30808™. The actual data are not provided and no details on experimental design and controls. For example, what concentrations were tested, how many technical and/or biological replicates, are the data means and standard deviations, what analysis method was used to calculate the IC50 and IC90, what is the Hill slope, what was set as the maximum activity control. The effective concentrations, termed as IC50 and IC90 in the manuscript, are surprisingly very similar (~1.6x and overlapping within the errors) which suggests a steep activity curve, but it is unclear due to lack of experimental data and details provided. These are important information as rest of manuscript and key conclusions are based on these data set.
§ To demonstrate selective activity against eukaryotic parasites over humans, the value of testing cytotoxicity in murine macrophages cell line J774. A1 (ATCC® TIB-67) is unclear. Even in this cell line, nitroxoline cytotoxicity was measured only for 24 hours, while anti-parasitic activity was evaluated with a longer exposure of 48 hours, making it difficult to directly compare selectivity. With all these caveats the demonstrated selectivity is rather very small of ~3.78-fold, which can also be concluded as a lack of selectivity within experimental error. Furthermore, the cytotoxicity assay also lacks important methodology, data and analysis details mentioned in the above point for the anti-parasitic assays.
§ Lines 109-116: Interestingly, the selectively of nitroxoline has been compared with miltefosine alone. A selectivity of 3.78 for nitroxoline and 3.3 for miltefosine for ATCC® 30808 over J774.A1 cells appear statistically similar within the biological and technical variability of the assays. An alternate interpretation of the data is that miltefosine is significantly less active but is also ~21-fold less cytotoxic (CC50 ~128 µM) than nitroxoline (CC50 ~6.15 µM). Given there is an urgent need for non-cytotoxic and more efficacious drug, nitroxoline does not seem to fit the need in terms of the in vitro data alone provided.
§ Lines 121-124: The cyst activity of 1.26 ± 0.42 µM and trophozoite activity of 1.63 ± 0.37 μM appear statistically similar within technical and biological assay error, and the former cannot be concluded as better. Also, what was the active control used for the cyst activity assay.
§ Lines 135-137 and 140-141 are incorrect, please rectify. Hoechst 33342 binds to all DNA, with condensed DNA having a higher fluorescence signal compared to rest of the DNA. It appears as if the microscope settings were set to focus on brighter signals from condensed DNA and filter out weaker signals. Kindly clarify.
§ Figure 1: Does not have a title.
§ Figures 1 to 5 state statistical analysis were performed using ANOVA. However, ANOVA is used for comparing three or more means but all these figures only have 2 means for comparison.
§ Line 144-147: ANOVA cannot be applied to these datasets as Hoechst and PI are different readouts, and by nature there can be differences amongst negative controls (untreated samples) for the different readouts which will bias the statical analysis.
§ Figure 1 and Figure 2: Can the authors please comment why the propidium iodide staining is significantly weaker than SYTOXTM Green dye even though both should be measuring the same.
§ Figures 1 to 6: Positive control not shown, and/or no existing drug(s) included for comparison to interpret the data in context of manuscript goals (as mentioned above).
§ Figure 6: Only 1 parasite is shown for each condition. Can the authors pleasure show quantification and an image with a larger number of parasites for representation as with other figures.
§ At the start of discussion, please summarize the data and findings.
§ Lines 266 to 268: The reference cited discusses 8-hydroxyquinolines in general and does not support the statement for nitroxoline, including ability to cross BBB.
§ Lines 285-286: It appears the statement is contradictory to the conclusions of the manuscript. If apoptosis is a survival mechanism by the parasite, then exposure to suboptimal doses of nitroxoline (~IC90 or ~1.6x IC50) could be a survival mechanism wherein some parasites self-destruct while others survive. There is evidence of this in Figures 1, 2 (and 3) wherein only 10-60% parasites seem to be affected even at IC90. Based on all the data it is unclear if nitroxoline has cidal activity against N. fowleri as some of the existing drugs like miltefosine. One way to evaluate this is to wash off the nitroxoline after 2 days of treatment and evaluate if parasites recover. Miltefosine (at 80 µM) can be used as a positive control in the experiment.
§ Nitroxoline is well known to induce apoptosis in vitro in human cells at low micromolar concentrations. An alternate interpretation of the data is that nitroxoline has a similar mode of action in Naegleria fowleri.
Reviewer 5 Report
The manuscript "Repurposing of Nitroxoline as an anti-Naegleria compound" is well described about the anti-Naegleria screening of an antibiotic Nitroxoline.
The introduction part is very informative. But it would be more appropriate to insert the structure of Nitroxoline and add some information on antibiotic properties of it.
In general, Primary amebic meningoencephalitis (PAM) is a disease which mainly affects the central nervous system. The records shows that it is a rare disease which is almost always fatal. Only 4 out of 157 people in the United States have survived from this infection during the period 1962 to 2022. Those survivals are early diagnosed the PAM and started the early antibiotic treatment only saved them. For the treatment of PAM, a combination of drugs which includes amphotericin B, azithromycin, fluconazole, rifampin, miltefosine, and dexamethasone are used. The authors did excellent work to evaluate the PAM activity of another antibiotic Nitroxoline. But it would be more informative to compare with the present using antibiotics. It would definitely increase the value of this manuscript. I suggest the authors that, if it is possible, repeat the experiments with Nitroxoline along with some other currently using antibiotics to compare the efficacy of the Nitroxoline in PAM treatment.
Also please mention the purchase details and purity of the Nitroxoline in Materials and Methods section.
Round 2
Reviewer 4 Report
The authors' revision of the text has improved the manuscript compared to the previous version. Thank you to the authors for sharing additional supplemental information to address some of my questions as well.
1. Recommend to add a Figure 1 with anti-parasitic activity and CC50: Since the key data of the study is nitroxoline's activity against the trophozoite and cysts stage, I recommend the authors add a Figure 1 (in place of current Figure 7) that shows the structure of the compound along with dose response activity against the two stages with the IC50 and IC90 mentioned in the figure (for example, above the dose response curve) - ideally all experiments combined data or at least a representative data set. The dose response data from multiple experiments can be combined by calculating percent inhibition for each drug concentration using DMSO alone as a control ( (measured RLU - DMSO RLU) / (DMSO RLU) ) x 100. A graph to demonstrate either CC50 or selectivity would also be meaningful. Kindly include in the figure legend that anti-parasitic activity was measured for 48 hours versus cytotoxicity measured for 24 hours.
2. Line 26, 28, 273 and 485: "high activity" is not clear, alternate suggestions include "low micromolar activity", or "low single-digit micromolar activity", or "potent anti-parasitic activity"
3. Line 29: Please expand PCD.
4. Line 78: Location of "8-hydroxy-quinoline" in brackets in the sentence is confusing as it is itself a derivative of quinoline, and nitroxoline is a 5-nitro-8-hydroxy-quinoline, which is a derivative of 8-hydroxy-quinoline.
5. Line 117: Nitroxoline has been tested against various cell lines for cytotoxicity in reference 31 Figure 3B and Table S3. These data are impactful (and advantageous) for the current study as Nitroxoline's CC50 varied from ~ 5 to 52 micromolar after 72 hours of drug exposure. This can be highlighted even in the abstract.
6. Line 132: Please expand PCD.
1. Line 131 and 132: Unclear sentence.
2. Line 134: "The" can be deleted before "Amphotericin B".
3. Line 292 and 293: Sentence at the start of the paragraph unclear.
